# In situ analysis of catalyst composition during gold catalyzed GaAs nanowire growth

Carina B. Maliakkal [1,2]*, Daniel Jacobsson [1,3], Marcus Tornberg [1,2], Axel R. Persson [1,3], Jonas Johansson [1,2], Reine Wallenberg[1,3] & Kimberly A. Dick [1,2,3]

Semiconductor nanowires offer the opportunity to incorporate novel structures and functionality into electronic and optoelectronic devices. A clear understanding of the nanowire growth mechanism is essential for well-controlled growth of structures with desired properties, but the understanding is currently limited by a lack of empirical measurements of important parameters during growth, such as catalyst particle composition. However, this is difficult to accurately determine by investigating post-growth. We report direct in situ measurement of the catalyst composition during nanowire growth for the first time. We study Au-seeded GaAs nanowires inside an electron microscope as they grow and measure the catalyst composition using X-ray energy dispersive spectroscopy. The Ga content in the catalyst during growth increases with both temperature and Ga precursor flux.

[1] NanoLund, Lund University, 22100 Lund, Sweden. [2] Solid State Physics, Lund University, Box 118, 22100 Lund, Sweden. [3] National Center for High Resolution Electron Microscopy and Centre for Analysis and Synthesis, Lund University, Box 124, 22100 Lund, Sweden. *email: carina_babu.maliakkal@ftf.lth.se

Nanowire growth by the vapor–liquid–solid (VLS) method is an important technique for producing well-controlled nanocrystals suitable for quantum components. For III–V semiconductors, an important material system for future technologies within electronics, solid-state lighting and quantum processing, VLS growth enables the fabrication of, for example, lattice-mismatched heterostructures[1–3], metastable crystal phases and crystal-phase tuning[4–7], and unusual ternary alloys[8,9]. VLS growth has been well-studied for over two decades, and extensive theoretical efforts exist to explain the growth process itself[10–14], the observed trends with experimental parameters[15–18], and the existence of metastable structures[19–21]. However, validation of theoretical predictions remains extremely difficult due to the large number of variable material properties and accessible experimental parameters, and the subsequent variance in reported experimental trends. In addition, many of the important fundamental parameters, such as surface and interface energies for relevant growth conditions, are unknown[22,23]. Consequently, there are a wide range of competing and complementary models that can explain observations, such as crystal phase trends[6,24–26] and diameter-growth rate dependencies[27–32].

The use of in situ characterization methods to gain direct insights into semiconductor nanowire growth in real time is one of the most effective ways to refine theoretical explanations and predictions, and in turn to better understand the conditions needed to design these materials with high control[33]. Examples of previous in situ studies include X-ray diffraction to understand phase and structural evolution[34–36], infrared spectroscopy to correlate surface chemistry during growth with resulting nanowire morphology[37–39], reflectance high-energy electron diffraction to follow nucleation and structural changes[40–42], optical reflectometry to monitor growth rate evolution[43,44], and mass spectrometry to study nucleation[45]. In addition, in situ scanning electron microscopy has been used to directly follow nanowire growth and morphology, and combined with Auger electron spectroscopy to track surface chemistry[46]. Finally, in situ transmission electron microscopy (TEM) has proven to be one of the most powerful ways to gain insight into nanowire growth in a directly interpretable way. Importantly, the information provided by this method is local, meaning that the nanowire, the growth-enabling liquid droplet and the interface between them can be visually identified and independently studied. This method has led to numerous significant breakthroughs in nanowire growth such as vapor–solid–solid growth[47,48], corner truncation[49,50], and step flow[48]. In situ TEM studies have been particularly beneficial for understanding bilayer growth kinetics[51], crystal-phase switching[52], triple-phase-line nucleation[53], and double-bilayer growth[54] in III–V nanowires.

One essential aspect that remains to be investigated is the local composition of the nanostructure during growth. This is necessary to understand composition evolution in, for instance, heterostructure and ternary nanowires, but even more importantly, for understanding the composition of the liquid metal droplet as a function of growth parameters and how this is correlated with the resulting nanowire properties[55]. The composition of the catalyst particle is a pivotal factor that determines its thermodynamic parameters, such as vapor pressure, chemical potential, and surface energies, which in turn decide the nanowire structure, growth rate, composition, etc.[56–58]. So far, the composition of the catalyst particles have been measured post growth and was shown to depend on the conditions used for terminating the growth[4,59,60] (more details in Supplementary Discussion 1), implying that post-growth composition of the particle is different from its composition during growth. To our knowledge, there has been no direct in situ measurement of catalyst composition during nanowire growth. An indirect estimation of the Au–Ga catalyst composition during GaAs nanowire growth has been reported by comparing the dimensions of the starting Au seeds particles and the catalyst during growth by assuming that the seed material does not diffuse out of the catalyst particle[52].

In this article, we report direct in situ measurement of catalyst composition during nanowire growth. We use in situ X-ray energy dispersive spectroscopy (XEDS) combined with in situ TEM to investigate the composition of the metal droplet as a function of growth parameters for Au-seeded GaAs nanowires grown by the VLS method. We show that the droplet consists of a significant quantity of Ga for all growth conditions, which increases with temperature for constant precursor flow. We do not observe any As significantly above the detection limit of the XEDS technique. Using calculated ternary-phase diagrams, we show that a lower bound on the As content can, however, be estimated. We also observe that the Ga content of the droplet for a given temperature is relatively independent of the ratio of V/III precursor species, so long as this ratio is above a certain threshold. Below this threshold, the Ga content increases strongly with decreasing V/III ratio, accompanied by a volume increase in the droplet. We show that the droplet volume scales with the Ga content, validating earlier works that used volume as an estimate of Ga[52]. The trend with V/III ratio is understood to correlate with a gradual transition toward the so-called "arsenic-limited" growth, whereby the droplet initially swells up but eventually stabilizes[61]. The results demonstrate that in situ XEDS is a useful and direct way to gain important insight and information on nanowire growth in real time, and will be similarly appropriate for other types of processes occurring at similar temperatures and overall gas pressures. Finally, the measurements of the droplet composition as a function of growth parameters will provide important inputs for validation and modification of theoretical models describing nanowire growth[14,55,62].

## Results and discussion

**Growth of GaAs nanowires and in situ measurements.** Au nanoparticles deposited on a silicon nitride-based heating grid were used to seed the nanowire growth. Nanowires were grown inside a Hitachi HF3300S environmental TEM integrated with a custom metal organic chemical vapor deposition (MOCVD) system. Trimethylgallium (TMGa) and arsine ($AsH_3$), which are the most common precursor gases in MOCVD growth of GaAs, were used for this study. The chemical composition of the catalyst was studied by XEDS as a function of temperature and the ratio of precursor fluxes, which are two very important parameters in typical MOCVD growth. Please refer to the Methods section for more technical details.

**XEDS of catalyst measured in situ.** The catalyst composition was measured using XEDS in situ as the GaAs nanowire grows. An example XEDS spectrum is shown in Fig. 1b from the nanowire shown in Fig. 1a. Clear signals from Ga and Au in the catalyst are observed for all spectra, along with a strong Si signal arising from the $SiN_x$ grid on which the nanowires are growing, and system artefacts such as Cu arising from microscope components. (The same spectrum but with a broader $x$-axis range is shown in Supplementary Fig. 1.) Quantification of the XEDS spectrum in Fig. 1b, measured at 440 °C, gives a Ga:Au atomic ratio of ~30:70 (assuming only Ga and Au are present). Some spectra also show small features that could be due to As, amounting to a maximum of ~1–3 atomic % (with normalization Au + Ga + As = 100%). However, the peak is too small to be conclusively attributed to As from the catalyst and quantified. Scattering into the GaAs nanowire could easily give rise to this signal (further information is available in the Methods section), in which case the Ga

concentration mentioned throughout the draft could be over-estimated by a couple of percent. Though any As within the catalyst is too low to be directly quantified, it is certainly below 3%. (So in the following sections, we quote Ga% and Au% measured in the catalyst after renormalizing such that Au + Ga = 100%, unless otherwise specified.) Our observation of very low concentration of As in the catalyst during growth is consistent with theoretical calculations; for instance, Glas et al. calculate the As content to be ~1% (depending on radius and contact angle)[63]. Mårtensson et al. predicted As content to be roughly in the range 0.01–1% depending on the growth conditions[64]. Post-growth XEDS reports have also claimed As to be below detection limits[60].

**Catalyst composition as a function of temperature**. The composition of a catalyst particle during nanowire growth was measured as a function of temperature and is shown in Fig. 2. During this experiment, the Ga precursor flow was set to be relatively low (V/III = 3780; the relevance of this choice will be discussed later).

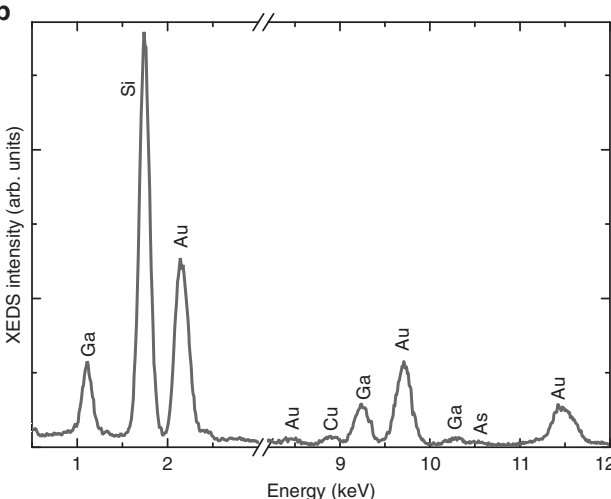

**Fig. 1** In situ Au-seeded GaAs nanowire growth. **a** TEM image of a GaAs nanowire growing inside the TEM on a SiN_x grid at 440 °C, and V/III ratio of 3780. Scale bar indicates 5 nm. **b** XEDS spectrum of the catalyst particle at the same conditions measured in situ during its growth. The atomic species giving rise to the different peaks are indicated in the plot

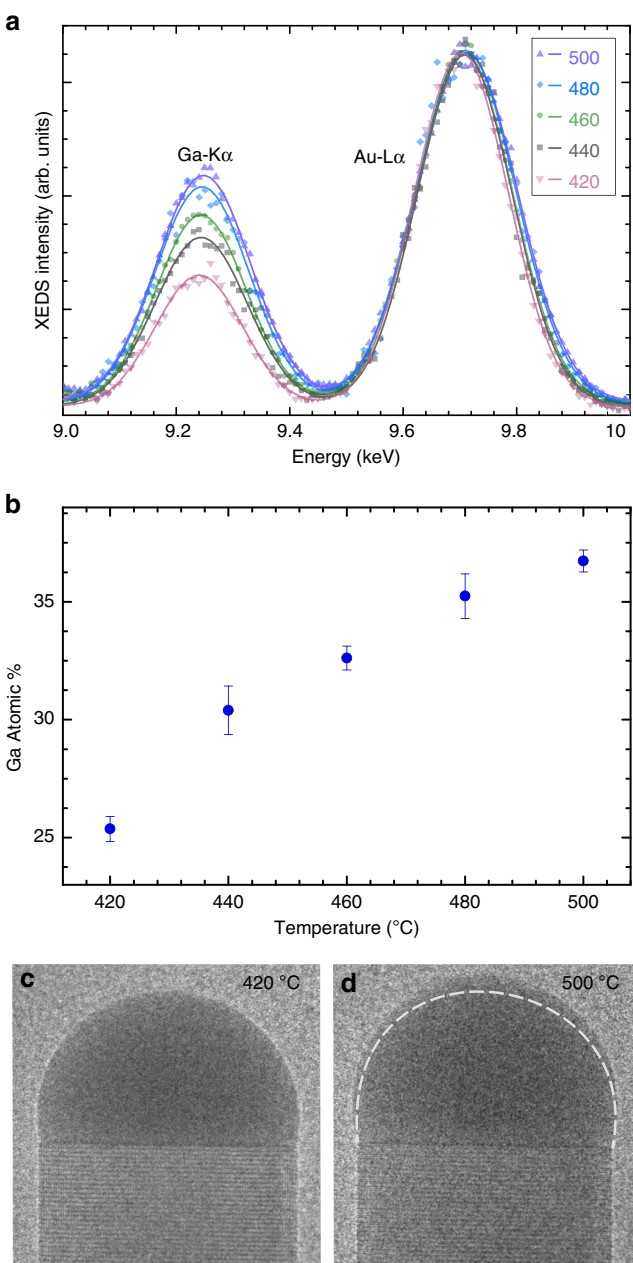

**Fig. 2** Catalyst composition as a function of temperature. **a** The Ga-K$\alpha$ and Au-L$\alpha$ peaks in XEDS spectra measured at different temperatures (in °C). These spectra are normalized with respect to the Au-L$\alpha$ peak. The Ga peak intensity increases relative to the Au peak with increasing temperature. **b** The atomic percentage of Ga in one catalyst particle measured as a function of temperature at constant V/III ratio. Error bars show SD in the XEDS quantification. With increasing temperature, the catalyst stabilizes with more Ga resulting in larger catalyst particle as seen in panels **c** and **d**. TEM images of the catalyst at 420 °C (**c**) and 500 °C (**d**). At 500 °C, the catalyst is larger than at 420 °C. The outline of the catalyst at 420 °C is depicted on top of the TEM image at 500 °C (**d**) by a white dashed line. Scale bars in panels **c** and **d** indicate 5 nm

More details of the experiment can be found in the Methods section. The Ga content in the catalyst increases with temperature as can be observed in the XEDS spectra normalized to the Au-L$\alpha$ peak (Fig. 2a). Quantification of the XEDS spectra shows that when the temperature was increased monotonically from 440 to 500 °C, the Ga content increased from 30 to 36 atomic % (with Au + Ga = 100%) (Fig. 2b). The temperature was then decreased to 420 °C, after which the Ga content decreased. A small but measurable change in the volume of the catalyst droplet at different temperatures was observed, as illustrated in Fig. 2c, indicating that the composition change is primarily due to an increase in Ga rather than a loss of Au atoms. (The correlation between Ga concentration and catalyst volume is discussed in more detail in Supplementary Discussion 9.) A separate experiment performed on a different nanowire, at a relatively lower temperature range than shown in Fig. 2 is given in Supplementary Fig. 3; which also shows an increase in the Ga content with temperature. The effect of temperature on growth is multifold—precursor decomposition (of both TMGa and AsH$_3$), Ga surface diffusion on the nanowire sidewalls, As evaporation rate, flow patterns in the growth cell, surface energies etc. depend on temperature. So a straightforward explanation of the observed trend is difficult.

The TMGa supply for this temperature series experiment was in a regime where the Ga content is rather insensitive to the Ga flow at 420 °C (will be explained later), indicating that it is not kinetically controlled. We therefore turn to thermodynamic considerations to understand the observations. Nanowire growth is understood to occur once the growth species becomes supersaturated in the catalyst[65,66]. Generally the growth is described as a cyclic process where a refilling step alternates with a layer growth step, and so the Ga and As concentrations oscillate between a maximum (just before a layer starts to grow) and a minimum (when a layer finishes). The lowest values for the minimum concentrations correspond to the equilibrium concentrations of these species in the liquid catalyst, termed the thermodynamic reference state of the system[66]. Ternary-phase diagrams provide a visualization of this reference state. The maximum allowed As and Ga content for a liquid Au–Ga–As alloy in thermodynamic equilibrium is given by the liquidus line (blue line in Fig. 3a) in the phase diagram. Since the catalyst must be supersaturated during growth, the composition must always be (slightly) to the right of the liquidus line, allowing us to estimate the minimum As concentration based on our measured Ga concentrations. Liquidus lines for a few representative As concentrations are shown in Fig. 3b. Using the measured Ga % at 420 °C and imposing the necessity of supersaturation, we can estimate the minimum As % in the catalyst to be ~0.01%.

It is worth noting that the Ga % measured by XEDS while the nanowire grows is the average concentration over a period of several minutes (typically 4 min), rather than the minimum Ga concentration. For a slightly lower Ga concentration, As would be slightly higher, and so the minimum As % of ~0.01% is a conservative order-of-magnitude estimate: the real value is expected to be slightly higher. An upper bound on the As concentration cannot be deduced from thermodynamic considerations, but is expected to be on the order of 1% based on the in situ XEDS measurements discussed above. Interestingly, the number of excess atoms required to form an entire GaAs bilayer would correspond to ~1% in the catalyst droplet for the catalyst/nanowire dimensions discussed here, meaning that there are never enough atoms stored in the droplet at one time for an entire layer to form at once. For the parameter ranges used in this article, it is indeed observed that the growth of each layer is not instantaneous[67]. (Further details on the step-flow propagation are discussed as a separate study in ref. [67], including experiments at conditions similar to that being discussed here.)

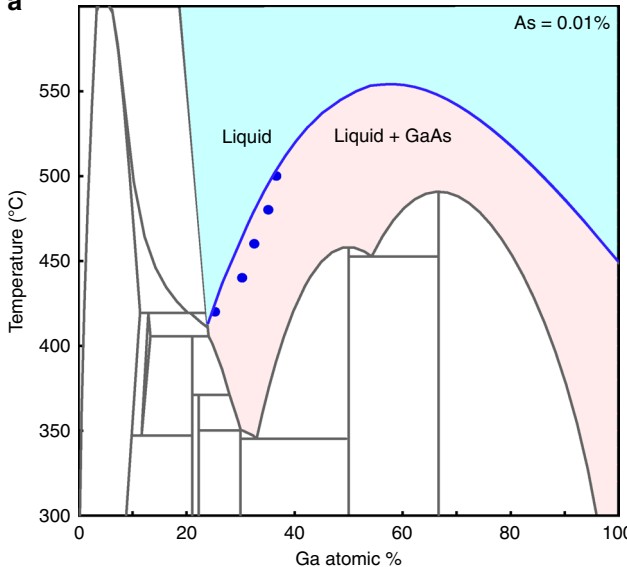

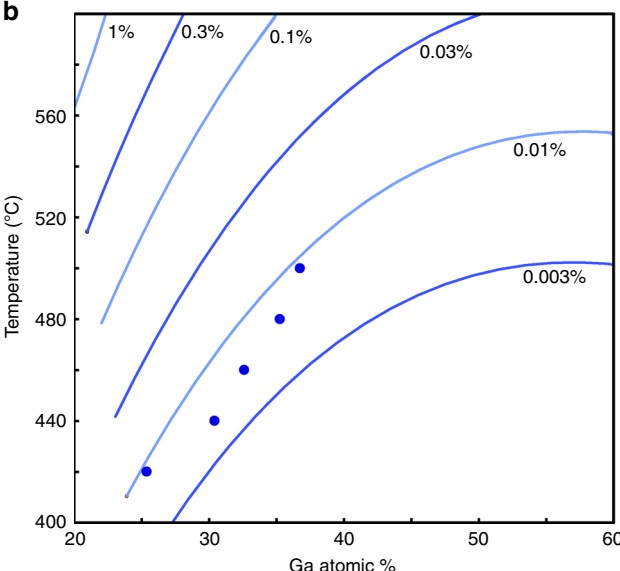

**Fig. 3** Comparison of measured Ga with constant As phase diagram sections. **a** Au$_{99.99\%}$As$_{0.01\%}$–Ga$_{99.99\%}$As$_{0.01\%}$ phase diagram (or projection of the Au–Ga–As phase diagram at a fixed As content of 0.01%). The choice of this concentration of As was such that the measured Ga percentage lies in the supersaturation regime (i.e., right side or below the liquidus line shown by the blue curve). Blue dots are the experimentally measured Ga content. **b** Liquidus line calculated for different As concentrations. The As concentration is labeled on each liquidus

**Catalyst composition as a function of precursor flux.** V/III ratio i.e., the ratio of the group V precursor flux (AsH$_3$ here) to the group III precursor (TMGa here), is a very important parameter for the growth of III–V semiconductors. We now discuss the change of catalyst composition as a function of V/III ratio, at a fixed temperature of 420 °C measured on another nanowire (Fig. 4a, b). (Please see the Methods section for details on how the V/III ratio is measured in this experiment). We had set the AsH$_3$ flow to be fixed and changed only the TMGa flow in this experiment. When the TMGa flow was stopped the nanowire was neither growing nor etching; the Ga content in the catalyst was then measured to be 27% by XEDS. (At elevated temperatures

GaAs gets slowly etched by the Au-based catalyst if the precursors are not supplied appropriately[4,68]). The Ga concentration in the catalyst then increases monotonically with increasing Ga

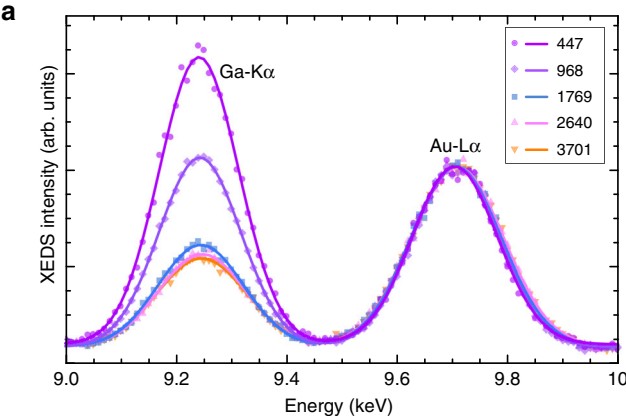

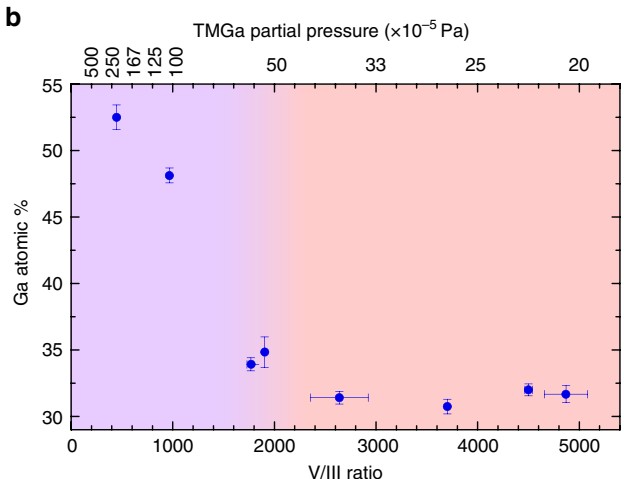

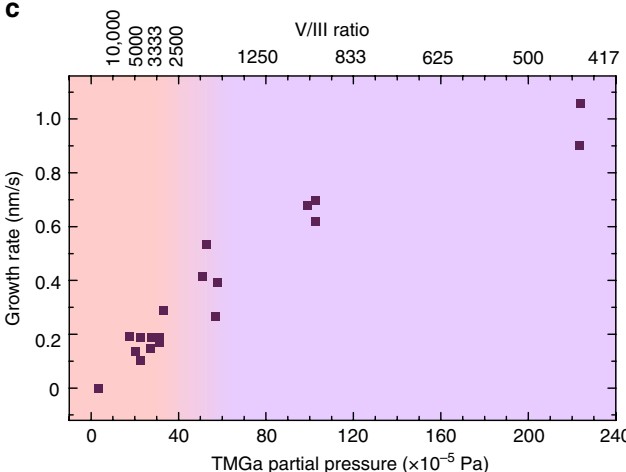

**Fig. 4** Catalyst composition and growth rate as a function of V/III ratio. **a** The Ga-K$\alpha$ and Au-L$\alpha$ peaks in XEDS spectra measured at different V/III ratios. These spectra are normalized with respect to the Au-L$\alpha$ peak. The Ga peak intensity relative to the Au peak increases with decreasing V/III ratio. **b** The Ga content in one catalyst particle measured for varying V/III ratio. Error bars show SD in the XEDS quantification. **c** Growth rate measured from the videos plotted as a function of TMGa partial pressure. The background gradient color in **b**, **c** is such that purple indicates high TMGa regime while peach indicates low TMGa regime. Note that the top axis in **b**, **c** is nonlinear

precursor flux. At high V/III ratios (or low TMGa) a small increase of TMGa does not change the Ga concentration much; in fact, the Ga concentration is effectively constant within the resolution limit of the XEDS measurement. Since the Ga content is in a steady state (between incoming Ga, controlled by the TMGa flow, and outgoing Ga, primarily controlled by the nanowire growth), this result suggests that the nanowire growth is mainly limited by the TMGa flow in this regime. This is the regime that we used for comparison with phase diagrams. At low V/III ratios i.e., below ~2000 (Fig. 4b), the Ga concentration increases rapidly with increasing TMGa. (A similar increase of Ga content at lower V/III ratio was observed at 500 °C by changing Ga flow. This is shown in Supplementary Fig. 5. An analogous trend was also observed when V/III ratio is varied by changing the AsH$_3$ flow instead, as shown in Supplementary Fig. 6). This trend in the measured composition by XEDS is accompanied by a large, clearly visible increase in the size of the catalyst droplet. Similar swelling of the catalyst at low V/III has previously been observed both ex situ[69,70] and in situ[51,52], and has also been predicted theoretically[71]. Previous reports have associated this effect with a transition to a V-limited regime where the Ga supply is effectively higher than the As supply, and so excess Ga accumulates in the droplet before reaching a new quasi-steady-state composition[51,52].

The growth rate of the nanowire as a function of TMGa partial pressure (and V/III ratio) is shown in Fig. 4c. It is clear that the growth rate increases with TMGa flow over the full range. The trend is not linear; however, for low TMGa (high V/III), there is a steep increase, but at higher TMGa (low V/III) this trend slows, potentially saturating at very high TMGa. The apparently linear trend between growth rate and TMGa for low TMGa flow is consistent with our interpretation above that the (effectively) constant Ga concentration in the catalyst is a consequence of Ga limiting the growth rate. Following the reasoning of Mårtensson et al.[64], we conclude that when AsH$_3$ is very much in excess, the As concentration in the catalyst quickly reaches a maximum concentration which is in steady state with re-evaporation to the vapor; so long as the nucleation barrier does not shift significantly with growth parameters, the growth rate is limited by the time required for the Ga concentration to reach the level needed to overcome the nucleation barrier. The weakening of this trend at high TMGa, where increased Ga is observed in the catalyst, indicates a transition to a growth regime where As plays a limiting role. In this regime, the high TMGa flow allows the Ga concentration to exceed the value reached in the high V/III regime, before the As reaches the concentration that would be in steady state with vapor. Since the supersaturation is determined by both Ga and As species, the nucleation barrier is then reached for higher Ga and lower As concentrations (determined by the V/III ratio). The growth rate will then depend on both TMGa and AsH$_3$ flows. True As-limited growth would be predicted for even higher TMGa flows, although it is not clear whether such a regime could actually be reached in experiments.

In addition to the change of the droplet size with V/III ratio, a change in the crystal structure of the nanowire was also observed. In the high V/III ratio regime (~2500–5000), the nanowire grew in the wurtzite structure along the ⟨0001⟩ direction (Fig. 5a, c). When the V/III ratio was decreased, stacking faults started to appear in the wurtzite nanowire. At even lower V/III ratios close to 1000, the nanowire grew as a mixture of both zincblende and wurtzite structures. At still lower V/III ratio of ~450, the same nanowire grew in the zincblende structure along the ⟨111⟩ axis (Fig. 5b, d). The change of nanowire structure from zincblende to wurtzite with increasing V/III has been reported experimentally[52,69] and

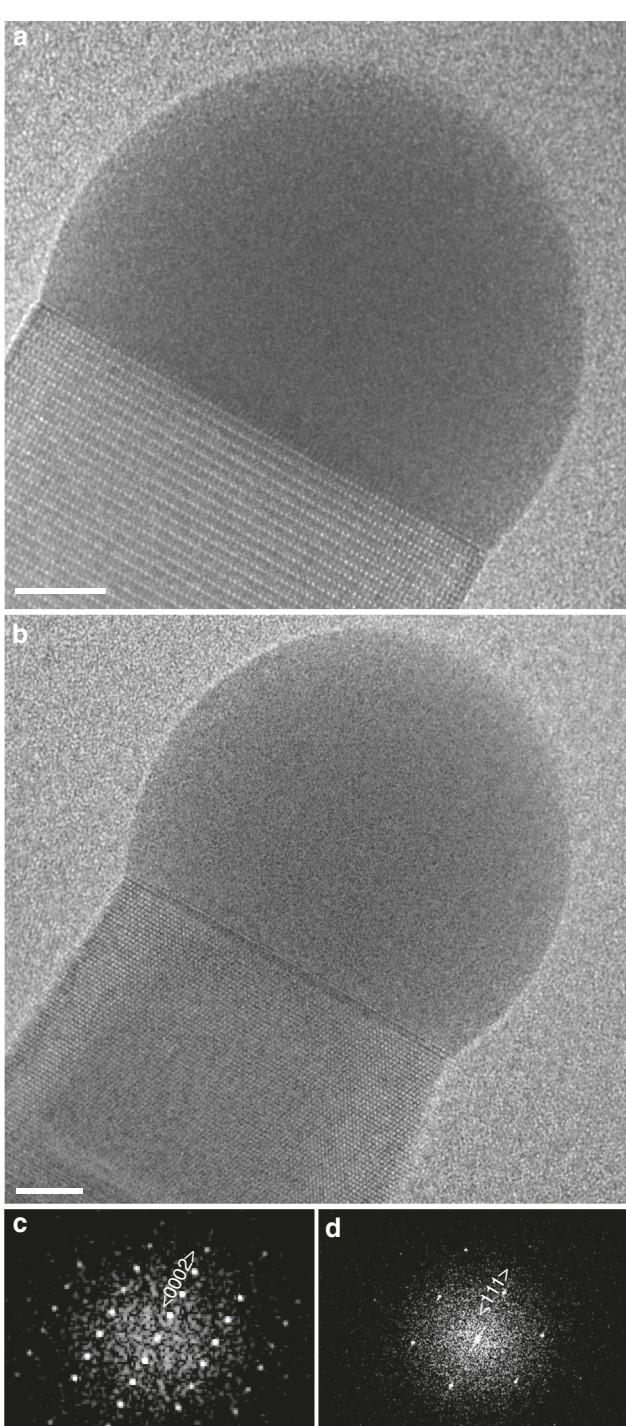

**Fig. 5** Catalyst morphology and nanowire structure. **a** TEM image at high V/III ratios where the nanowire is growing in the wurtzite structure. **b** TEM image at low V/III ratios where the nanowire is growing in the zincblende structure. **c**, **d** are processed reduced FFTs of panels **a** and **b**, respectively, showing wurtzite and zincblende structure, respectively. Scale bars in panels **a** and **b** denote 5 nm. Scale bars in panels **c** and **d** indicate 5 nm⁻¹.

theoretically[24]. MOCVD experiments also show a second transition back to zincblende at very high V/III ratio[64,69], but this regime was not covered in the experiments described here. We have also observed in other experiments that, for even higher TMGa flows than studied here, the interface develops an

oscillating truncation consistent with earlier in situ observations[49–52]. Since the interface dynamics are qualitatively very different in that regime, it was not covered during the experiments that are included in this study.

## Summary and outlook

In summary, GaAs nanowires were grown with a gold-based catalyst particle in an environmental TEM in order to deepen the understanding of nanowire growth. The chemical composition of the catalyst particle was measured in situ as the nanowire was growing. We report the catalyst composition during growth of one nanowire as a function of temperature and another wire of similar dimensions as a function of ratio of precursor fluxes. Since the Ga concentration in the Ga-limited growth regime seem to be determined by thermodynamics, the results would be applicable to Au-catalyzed growth of GaAs nanowires independent of the growth method. Although the As content in the catalyst is close to the detection limit by XEDS, we can estimate for the As concentration a lower bound (by comparing measured Ga–Au content with calculated Au–Ga–As phase diagrams) and an upper bound (from XEDS). The Ga concentration in the catalyst increases with increasing Ga precursor flux. These in situ measurements will aid better theoretical modeling of nanowire growth and improve the understanding of nanowire growth mechanisms. The precursor partial pressures and the nanowire growth rates in these experiments are similar to that in a typical ex situ MOCVD, which is why we believe the quantitative results will be applicable to the latter. The results could be extended at least qualitatively to other growth techniques as well. Most metal-assisted III–V and II–VI nanowire growths typically have low solubility of the anion in the catalyst, and the Au-assisted GaAs studied here serves as a model system.

## Methods

**In situ nanowire growth**. Gold aerosol particles of 30 -nm diameter on an average were used to seed nanowire growth. Silicon nitride MEMS heating chips from Norcada were used as the substrate. The thinnest SiN$_x$ parts where growth was monitored had a thickness of ~35 nm. Atomic-resolved imaging was performed with an AMT XR401 sCMOS camera, and the videos were recorded at about 20 fps. The TEM images in the article were extracted from these videos and processed.

GaAs nanowires were grown in a Hitachi HF3300S environmental transmission electron microscope (ETEM) with CEOS B-COR-aberration-corrector and a cold-field emission gun. Blaze software by Hitachi was used to control the local sample temperature using Joule heating in a constant power mode. The chips are calibrated to the melting point of gold with accuracy of ± 5 °C by the manufacturer. The ETEM was connected to a gas handling system with the CVD gases. A single-tilt holder that has two separate microtubes running to the holder tip was used for supplying gases. The holder and the gas-handling system are connected by a polymer-coated thin quartz tube (PEEKSil) from Trajan Scientific.

**XEDS**. The XEDS measurements were performed with an SDD X-Max$^N$ 80T system from Oxford Instruments. While measuring the catalyst composition, a small condenser aperture was used and the beam was condensed on the anterior part of the catalyst (opposite to the nanowire/catalyst interface). As the nanowire grows, the catalyst particle keeps moving forward. During XEDS, the illuminated area on the fluorescent screen was continuously monitored and the sample moved appropriately. Electron dose was in the order of 20,000 electrons Å⁻² s⁻¹ during XEDS. We specify the percentage content of the elements in terms of atomic percentage (and not as weight percentage) throughout this paper. The data were acquired and quantified with Aztec software. Acquisition was typically for 4 min. In XEDS scans, we observe signals of Au, Ga, and As from the catalyst and/or nanowire, Si and N from the substrate and Cu due to scattering from parts of the microscope. The quantification result is renormalized such that Au + Ga = 100 atomic % to obtain the results quoted in this article, unless otherwise specified. The default lines (K for Ga, L for Au, K for As) were used for quantification. The standard deviation of quantification result from Aztec is shown as the y-axis error in Figs. 2b and 4b. The background subtraction of XEDS spectrum is done by filtered least-squares fitting. More details about the XEDS quantification results are given in Supplementary Discussion 6. In all the plots in this article indicating error

bars, the value is indicated as mean ± SD. The reliability of the XEDS quantification was cross-checked by measuring stoichiometric GaAs with the same instrument and quantification tools.

**Arsenic content from XEDS**. Typically, the measured As signal is very low, and a clear peak cannot necessarily be distinguished from background. Full quantification of spectra (including background and artefacts such as Si, N, etc.) yields an As weight % of less than one, which is close to the detection limit in XEDS. Moreover, any small As signal detected may not necessarily originate from the catalyst nanoparticle. Since the nanowire is growing during the XEDS acquisition, the catalyst/nanowire interface moves. In order to track this motion manually, we try to not condense the beam to a spot, but slightly spread it so that a small portion at the front rim of the catalyst is visible. Imperfections in tracking the motion of the particle can contribute to XEDS signal from the nanowire part. This along with electrons scattered into the nanowire from the catalyst and also the electron in the periphery of the direct beam could easily lead to an overestimate of both Ga and As by a few percent.

**Temperature series experiment**. In the temperature series experiment discussed, temperature was increased from 440 to 500 °C in steps of 20 °C, and finally decreased to 420 °C. After each temperature was reached, we waited at least a minute so that the catalyst stabilizes and there is no evident change in its dimensions. This wire grew in the wurtzite structure in the ⟨0001⟩ direction at the conditions studied.

Phase diagrams shown here are calculated with the Thermo-Calc software using the thermodynamic data assessed by Ghasemi et al.[66].

**V/III series experiment**. The measurement for the V/III series were recorded at 420 °C. We started to observe the nanowire when the V/III was 4500, where we measured the first XEDS data. Then the Ga supply was stopped for some time. The TMGa flow was restarted, and increased slowly in steps until (at very high flow) the nanowire changed direction and folded back on itself during an XEDS acquisition. During the XEDS measurement at the lowest V/III ratio (447), there was some As signal observed due to scattering from GaAs NW, so the same percentage of Ga was subtracted from the quantification results and renormalized to obtain the data point plotted (see Supplementary Discussion 6 for details). The entire range (not SD) about which V/III varied during individual XEDS spectrum acquisition is denoted by the error bar for the x-axis in Fig. 4b. Each point in the plot of growth rate (Fig. 4c) is one measurement of the average growth rate of the nanowire in a particular time interval. From two different frames of the recorded videos, the time difference between the frames and the length the wire has grown in that time interval is noted, and growth rate calculated. The time difference we chose was typically 10–20 s. In this interval, the variation/fluctuation in the precursor flow is less than a percent. So the error bars on the x-axis will be smaller than the symbols used.

**Pressure at the sample**. During growth experiments, the pressure near the pole piece was measured by an Inficon MPG400 pressure gauge, and is referred to as "column pressure" here. The precursor inlet tubes run along the length of the TEM holder, and precursor gases are released close to the heated $SiN_x$ grid. Hence, the pressures are higher at the growth front than the "column pressure". The sample pressure relative to the column pressure was calibrated using the pressure at the heating coil of a clean $SiN_x$ grid (without Au or GaAs) as a pressure gauge following the Pirani gauge principles using the Blaze software. We performed calibration experiments with $N_2$ and also $H_2$ and found that the pressure at the sample (measured by Blaze) is twice of "column pressure". A factor of two is therefore used to estimate sample pressure for each species based on its calibrated column pressure.

**Precursor partial pressures and V/III ratio**. The TMGa bubbler was maintained at −10 °C with $H_2$ bubbled through it. In addition, a small fixed amount of $H_2$ dilution (to be more precise, four times the flow used for the bubbler) was added in the TMGa line followed the bubbler. No additional carrier gas was used, and the $H_2$ partial pressure is thus much lower than in a typical MOCVD. The flow of the TMGa/$H_2$ mixture was controlled by a mass flow controller (MFC), and a portion of the resulting flow was bypassed to the vent line to restrict the TMGa pressure reaching the microscope.

In order to determine the partial pressure of TMGa at the sample, the precursor fluxes sent to the ETEM were monitored with a residual gas analyzer (SRS RGA 300) using mass spectrometry in these experiments. The amount of the dominant TMGa derivatives (containing Ga) are measured at mass-to-charge ratio of 101, 99, 71, and 69 associated with dimethylgallium and Ga. (The sample heating is very local at the $SiN_x$ grid, decomposing just a very small fraction of the supplied precursors and so these RGA measurements are independent of localized pyrolysis at the sample.) Calibration experiments were performed for different but known TMGa and $H_2$ flows to find the correlation between "column pressure" and the Ga-related mass spectrometry reading at mass-to-charge ratio of 101. We assume that

TMGa and $H_2$ are being pumped out from the ETEM at the same efficiency, which might not be true, and hence the TMGa partial pressure quoted in this report could be an underestimate. TMGa partial pressure at the sample during experiments is therefore determined using the calibrated RGA readings together with the factor of two between column pressure and sample pressure described above.

The $AsH_3$ flow was controlled exclusively by MFC (no part bypassed) and fixed for all experiments reported here at ~1 Pa at the sample. V/III ratio is then calculated using this value divided by the TMGa pressure calculated using RGA readings for the specific experiment.

Inficon MPG400 pressure gauge, that was used during growth experiments, is gas dependent. So using a gas-independent capacitance pressure gauge SKY$^R$ CDG045D, we found the correction factors for $H_2$, $N_2$, and $AsH_3$ in the appropriate pressure ranges. These correction factors were included in the precursor partial pressure calibration curves.

## Data availability

More data that supports the findings of this study are available from the corresponding author upon request.

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

## Acknowledgements

We wish to acknowledge support from the Knut and Alice Wallenberg Foundation, NanoLund, and the Swedish Research Council. We thank E. Mårtensson for valuable discussions and insights. We are grateful to Joacim Gustafsson, Stas Dogel, and Charles Soong from Hitachi High-Technologies and John Wheeler and Shaun Ohman from Collabratech for technical assistance in setting up the laboratory. We thank Robin Sjökvist for assistance during the precursor flow calibration experiments. Open access funding provided by Lund University.

## Author contributions

C.B.M., D.J., M.T. and A.R.P. performed the experiments. Data analysis was done by C.B.M. J.J. performed the phase diagram calculations. K.A.D. and R.W. coordinated the project. All authors discussed the results and contributed to the concepts discussed in this article.

## Competing interests

The authors declare no competing interests.
