## [Peer Review File · Nature Communications]

Response letter to reviewers' comments- NCOMMS-19-07494

We sincerely thank the reviewers for their evaluation of the manuscript and contributing to its improvement. The reviewers' comments are in black font and our responses are in blue italics. We have numbered the comments.

Reviewer #1 (Remarks to the Author):

This manuscript reports on the chemical composition analysis of the nanoscale Au droplets (named catalyst from here on) used for III-V nanowire growth. This kind of measurement is a premiere, as this high-end microscopes did not exist till very recently. Having direct access on the catalyst composition is very important for the understanding of the nanowire growth process. More than the group III composition is the role of the group V element that has intrigued scientists. Group V elements such as As is insoluble in Au, several hypothesis have been given on how Group V contributes to growth (diffusion through then liquid droplet or directly at the liquid-solid interface). By backing the compositional data with simulations, authors provide an estimation of the group V composition in the droplet. This finding is really new and relevant, it should be more highlighted.

We thank the referee for the appreciation and his/her effort towards improving the manuscript.

a) Authors should make the connection with ex-situ measurements, as this is the data that exists till today. In particular, it would be useful to know the catalyst composition and droplet shape upon cooling. This would answer the question of how different measurements are if they are taken in situ or after growth ex situ.

The catalyst composition after cooling is strongly dependent on the cooling conditions (such as the ambient gas and cooling rate). A few representative examples from literature on post-growth composition is shown in Supplementary Information (Table S1). (The cooling process will be a function of the instrument geometry, heating mechanism etc. making it difficult to replicate a particular strategy.)

Typical reported values of catalyst composition measured post growth when the nanowires where cooled down in an AsH₃ environment has a relatively less spread and falls in the 0-3% range. We attempted cooling the nanowires in AsH₃ following in-situ growth experiments, and found about 3-4% of Ga in the catalyst, tallying with earlier reports. We have now mentioned this in Table S1.

Fig. R1: TEM image of a catalyst particle after growth and cool down.

The composition reported in literature when the nanowires are cooled in a H₂ ambient is spread among a very broad range of values (Ref. Table S1), generally attributed to the specifics of the cooling process; for this reason, it would not be the best cooling method to compare with in situ experiments.

b) The composition of the catalyst is measured as a function of temperature. I believe the Ga content increases at high temperature because of the increased decomposition rate of the precursor.

In addition, the increase of temperature allows a much broader range of compositions to be achieved. The interplay between the chemistry and the thermodynamics is not clearly stated in the manuscript.

*We thank the reviewer for pointing this out. We agree to the reviewer's view and have **modified** the draft (page 6) appropriately.*

The effect of temperature on growth is multi-fold – precursor decomposition (of TMGa and AsH₃), Ga surface diffusion on the nanowire sidewalls, As evaporation rate, flow patterns in the growth cell, surface energies etc. depend on temperature. However, it is almost impossible to obtain numerical values for all the separate factors. Temperature increases AsH₃ cracking also in a very similar way to TMGa cracking,¹ so the effective V/III ratio might turn out to be similar at the studied temperatures; in such a scenario the change in catalyst composition would not be due to difference in thermal decomposition. But we agree to the reviewer that the possible impact from thermal decomposition cannot be completely ignored. Therefore, we have revised the manuscript to considering reduce the role of phase diagrams in interpreting the temperature series data.

In the modified manuscript, we are primarily employing phase diagrams to interpret the Ga concentration measured at 420 °C. The justification is as follows: The choice of the TMGa flow (2.6×10^{-4} Pa, V/III=3780) was such that it falls in the regime where Ga content in the catalyst did not change with V/III ratio plot at 420 °C (Fig. 4 b). In such a regime, the catalyst composition seemed to be determined by thermodynamics and not kinetics. (The TMGa value chosen for the temperature series was not in the V/III-insensitive regime for the 500 °C case, which is why we are not interpreting Ga % in the temperature series for values other than at 420 °C on the basis of thermodynamics and phase diagrams now.)

*We now **removed** the claim that the calculated phase diagram matches the observed temperature series data. However, since the Ga concentration at 420 °C is still decided by thermodynamics, we use the measured Ga content to obtain a lower bound on the As in the catalyst. We now **added** to the paper the several possible effects of temperature and that these cannot be separately and quantitatively assessed.*

That said, in a recent publication from our group where a Au-catalyzed NW is heated up in vacuum the Ga content observed to increase with temperature.² This might be an indication that the observed composition vs temperature trend has a thermodynamic contribution.

c) Moreover, the degree of confidence in the values of temperature are not given.

*We have now **added** the following sentence in the Methods sections. ‘The chips are calibrated to the melting point of gold with accuracy of $\pm 5^\circ\text{C}$ by manufacturer.’*

The SiN_x chips and heating software & hardware are from Norcada inc. and Hitachi High-Technologies. Simulations on the grid heating were performed and the heater settings required for reaching different temperatures were calculated. To cross-check, melting temperature (set point) of gold was measured experimentally from which the temperature accuracy was found to be $\pm 5^\circ\text{C}$. Au melting point is 1064 °C. The GaAs growth experiments discussed in this manuscript were in a different temperature regime (420-500 °C). The thermal balance may be different for temperatures below 700-750 °C and above, since the mechanism of thermal transfer changes from heat conductance to IR radiation. So the error bars could be in principle different, but we or the manufactures do not have any numerical estimates for temperature accuracy in the relevant regime.

d) Error bars are missing in Fig. 5c.

(Note that in the revised manuscript the figure that was Fig. 5(c) in the original manuscript is now 4(c))

We apologise for the confusing phrasing, each point is one measurement of the average growth rate in a short time interval of the same nanowire. We meant it was time-averaged over several seconds, and not for just one layer to grow. So there are no significant y-axis error bars arising from averaging multiple measurements (note also that measurement error of the length and time

are extremely small compared to the absolute values, much smaller than the size of the symbols used in the graph). From two different frames of the recorded videos the time difference between those frames and the length the wire has grown in that time interval is noted. The time difference we chose was typically 10-20 seconds. In this interval the variation/fluctuation in the precursor flow is less than a percent. So the x-error bars would also be smaller than the symbols used. We have now **added** a brief discussion of this process to the Methods section. We have **edited** the manuscript to read 'growth rate' instead of 'average growth rate'.

e) Authors could also work on the graphics. For example, figure 3 could be rendered much more visual and compact. I believe information on Fig. 3c could be added to the two above to render the information easier to read.

(This comment is mainly about the isothermal phase diagrams.) We have removed that figure from the main manuscript; part of it is shown in modified form in Supplementary Information (Fig. S4).

We have **modified** the format of the other images slightly in arrangement, colours etc.

f) Finally, authors have scanned only a limited range of growth conditions. Being able to perform in situ measurements and change the conditions in a fast manner, it should be possible to scan a broader set of growth conditions to back up the generality of the theory provided here and for the usefulness of the data to the scientific community working on this area.

We appreciate the reviewer's interest in more experimental parameters, and while we suspect they are overestimating the easy of conducting in-situ growth experiments, we have now **added** more experimental data for other growth conditions. Together with the temperature series from 420-500 C in Figure 2a, the V/III ratio series conducted at 420 C covering an order of magnitude of Ga pressure in Figure 4a and the second V/III ratio series covering a range of higher ratios at 500 C, we believe this now covers very well the parameter range for GaAs nanowire growth.

(i) We have now **added** another temperature series XEDS measurement for a different range of temperatures than was already in the manuscript to the supplementary information (section S4).

Fig. R2: Ga content in a catalyst studied as a function of temperature with fixed precursor flows.

(ii) Shown below is an XEDS series as a function of V/III ratio where the AsH_3 flow is changed by bypassing some of the flux passing through the MFC. The trend is similar to what we observed for series where TMGa was varied (Fig. 4 b). We have **added** this plot to supplementary information.

Fig. R3: Ga content in a catalyst with different precursor flows.

(iii) The V/III ratio measured at 500 °C, is shown in the supplementary information.

Reviewer #2 (Remarks to the Author):

This work presents a detailed analysis of the growth mechanism of individual gold-seeded GaAs nanowires. The characterisation of the growth parameters such as the catalyst particle composition is carried out in situ by means of electron microscopy measurements during growth. The thermodynamics of the growth processes are studied as a function of a number of factors including the temperature and the Ga precursor flux, and the results for the measured catalyst composition can then be directly compared with the theoretical predictions for the Au-Ga-As phase diagram.

The results of this paper are highly innovative, well presented, and certainly suitable to be published in Nature Communications. This study sheds unique light upon so far poorly understood aspects of the growth of GaAs nanowires, allowing us a deeper understanding of the underlying dynamics which in turn should allow to design and realise more complex nanowire-based heterostructures. This paper thus opens a new dimension for the studies of nanowire growth dynamics, and are thus certainly of direct relevance for this area of research.

We thank the reviewer for the appreciation.

Reviewer #3 (Remarks to the Author):

The manuscript describes growth of a gallium arsenide (GaAs) nanowire during chemical vapor deposition (CVD) onto a gold (Au) catalyst in-situ a transmission electron microscope (TEM). The main conclusion is that over a range of growth temperatures below $\sim 500^\circ\text{C}$, growth would be limited by the partial pressure of the Ga containing metal-organic precursor. This finding is based on measurements of the local chemistry by energy-dispersive x-ray (EDX) spectroscopy combined with an exploration of the Au-Ga-As phase diagram.

The manuscript is well written and the explanations are carefully worded.

We thank the reviewer for the constructive criticisms and suggestions. We have addressed each comment below.

This reviewer, however, sees a number of critical issues concerning the scientific reasoning and the interpretation of the actual measurements as well as the general choice of the journal.

a) Concerning the latter, the topic is narrow in scope and it remains highly questionable whether this (well explored and understood) materials system in combination with a specific type of growth within an even more highly specific environment can really serve as a general model system as suggested. The novelty really relates not to the materials system, as can be seen from the large number of references on prior art cited, including more than 20 earlier studies by the authors, but more to the way of implementing the experiments by EDX in an environmental TEM. Also, the manuscript of 20 pages length and an additional 5 pages supplement appears overly long for a communications or letter style contribution. It would probably be much more suitable for a regular contribution in a growth and/or microscopy method focused journal.

*We have **edited** the revised manuscript to address the reviewer's comments regarding novelty, length and implications to typical ex situ growths.*

*Though the importance of catalyst composition for nanowire growth is known in the nanowire community, a direct in situ measurement of catalyst composition during nanowire growth has never been reported in literature for any material system. We agree that nanowires are by no means a novelty, as there are several thousands of publications related to nanowires; but still the growth mechanisms are not yet clearly understood. On the contrary, we would argue that the very widespread interest in nanowires indicates the probably interest of this study to a broad community. The importance of this work is the direct measurement of catalyst composition, which critically affects the nanowire growth mechanisms. We have **revised** the manuscript to ensure that novelty and significance are clear to the reader. We believe this fits very well within the scope of Nature Communications as described as "research of interest to that specific research community".*

The revised manuscript with less than 3700 words and 5 figures fits well within the Nature Communications guidelines of up to 5000 words and up to 10 figures.

*This manuscript is about MOCVD growth of Au-catalysed GaAs nanowires studied in a very specialized environment, as mentioned by the reviewer. **Added** this to the summary - "The*

precursor partial pressures and the nanowire growth rates in these experiments are similar to that in a typical (ex situ) MOCVD, which is why we believe the quantitative results will be applicable to the latter. The results could be extended at least qualitatively to other growth techniques as well."

b) The main approach taken, combining EDX with calculated phase diagrams, does seem attractive at first sight but ignores two fundamental problems, one related to the validity of the phase diagrams for the set-up and the second related to x-ray analysis of the specific system investigated.

Both these points are discussed later in this document. But in short, we have now modified the interpretations where phase diagrams are used (details in comment c and d). We have also addressed the concerns about XEDS quantification (details in comments e and g).

c) The authors' statement on the relevance of phase diagrams to understanding nanowire growth (end of abstract) directly contradicts latter statements on page 2 where they acknowledge that equilibrium phase diagrams are perhaps invalid for understanding the non-equilibrium nanowire growth and on page 3 where they admit that nanowire growths is 'of course not' an equilibrium process. Herein they seem to refer to nanowire growth in general, but the problem is exacerbated here because, firstly, the pressure range that can be covered in gas-source growth in an environmental TEM is very different from that typically employed in large chamber CVD growth and, furthermore, electron beam irradiation will move the system even further away from equilibrium.

*We apologize for the confusing phrasing. Overall, we have cut down the discussions regarding phase diagrams in the manuscript and tried to clarify how we have used the phase diagrams as a lower bound on Ga and As%. (The abstract, main text and conclusion are all **modified** accordingly.)*

We summarize our views about usage of phase diagrams for nanowire growth here. Nanowire growth is not an equilibrium process. Phase diagrams describe systems under equilibrium, which serve as the reference state for the condition under which crystals grow. During a GaAs nanowire growth experiment, the Au-Ga-As catalyst particle would either be at thermodynamic equilibrium (immediately following formation of a layer, before atoms start to build up in the droplet again) or at higher Ga and As concentration than equilibrium (assuming number of Au atoms remain constant). Hence the phase diagrams provide a lower limit for the Ga and As concentration. In the revised manuscript we have limited our discussion of the measured Ga concentration combined with phase diagrams only to find a lower bound on the As concentration.

Regarding pressure: The pressure dependence of phase diagrams including only condensed phases, such as the Au-Ga-As diagram, is generally negligible for the pressures considered here. The underlying Gibbs free energies of the involved condensed phases are modelled as pressure independent.³ Moreover, though the overall pressure of the system is orders of magnitude different between the growth in an ETEM and a typical MOCVD reactor, the partial pressures of reactants and growth rates are similar.⁴ The (total) pressure in these experiments are higher than other in situ reports of MOCVD nanowires grown in TEMs,⁵ hence one step closer to a typical MOCVD.

*Beam effect: In the modified interpretation, we are taking into account that there could be more Ga and As in the catalyst than given by the equilibrium phase diagrams. Even if the electron beam is pushing the system away from equilibrium, the interpretations using phase diagrams still hold as we are only extracting a lower bound for As% now in the manuscript. We have **edited** to make this clear (page 6) (If the Ga% is lower at equilibrium that would only increase the corresponding value of As%.) Electron dose was in the order of 20,000 electrons/Å²s during XEDS, we **have added this** now to the Methods section.*

d) 'Whether phase diagrams like e.g. by Predel & Stein, Acta Met 20 (1973) 681 for the case of Au-Ga may really be applicable here is therefore highly questionable. It appears interesting to compare observations to such phase diagrams but under no account should they be used a-priori to draw any conclusions from. Two excellent papers on this topic that should be considered by the authors are Wang, He, Liu, Chong and Chen, Angew Chem Int Ed 54 (2015) 2022 on nanowire growth and Walther, J Microsc 257 (2015) 87 on in-situ TEM. Both warn on the limited usefulness of phase diagrams when extracting thermodynamic rather than kinetic data under non-equilibrium growth conditions.

The interpretations using phase diagrams are modified now, and considerably reduced in length, as elaborated in the previous point. In short, the catalyst composition during growth will have more As and Ga than equilibrium. If the Ga was lower than what we measured, then the corresponding As content from phase diagram will be even higher. Hence we are currently using the phase diagram only to find a lower bound on the Arsenic content.

As far as the knocking out of atoms by the electron beam is concerned, like what is discussed by Walther J Microsc 257 (2015) 87,⁶ then at steady growth parameters, the catalyst would shrink with progressing time. In our experiments, we have not seen this shrinkage even with several minutes (about 20 minutes) of waiting. Secondly, by comparing the volume change of catalyst to the measured composition (Fig. S7) we find that no significant portion of Au has left the catalyst during the experiment. Au atoms have more scattering cross section than Ga which makes Au more susceptible to the kicking out events. Also Au is deposited initially, to seed/start the growth (and not continuously supplied during growth). So the two observations (a) catalyst not shrinking with time and (b) no measurable Au diffusion out of the catalyst, implies that changes in composition due to knocking out of atoms is much smaller compared to the other measurement/quantification errors.

Wang et al.⁷ nicely explains that one should identify if the observations are kinetically or thermodynamically controlled. Our point is that, if the catalyst composition is controlled by kinetics and not thermodynamics, the measured Ga value can only be higher than that at equilibrium. (If there was less Ga than thermodynamic equilibrium, there would be no super-saturation, and hence no nanowire growth.) If the actual Ga content is less than what we measured, that would make the estimate of As% lower bound using phase diagrams (Fig. 3b) even higher, which will not be contradicting our current statement.

e) The second issue relates to the x-ray spectra of Au-Ga-As. Figure 1b, although not a complete

EDX spectrum, demonstrates that there is significant overlap between the x-ray lines of As K-alpha (10.5keV), Ga K-beta (10.3keV), Au L-alpha (9.7keV) and Au L-beta (11.4-11.6 keV). This makes detection of As in the presence of strong Ga and Au peaks, as is the case here, almost impossible given the weak As lines will be buried in the shoulders of much stronger peaks either side.

Regarding XEDS peaks and energy resolution:

We use a SDD X-Max^N 80 T detector for XEDS which has an energy resolution of 155 eV (or better), which is sufficient for resolving the Au L- α 1 (9.7keV), Ga K β (10.3keV), As K- α (10.5keV) and Au L- β (11.4 & -11.6 keV), peaks. Figure below (Fig. R4) is a zoomed in plot of the Fig. 1 (b) of the manuscript, showing that these peaks are well separated. However, the As K- α (10.5keV) and Ga K- β (10.3keV) peaks are somewhat close to each other. Since Ga K- β (10.3keV) is a relatively low intensity peak of Ga, the As K- α (10.5keV) can still be observed. If part of the As K- α (10.5keV) signal is wrongly interpreted during quantification to be due to Ga, the Ga quantification will not be significantly different as the Ga K- α at 9.2 keV peak is much stronger. If part of the Ga K- β (10.3keV) signal is wrongly calculated as As, then As will be slightly overestimated. Arsenic XEDS quantification data is interpreted as only an upper limit for As% in the catalyst, so the claims in the paper will still be valid.

Fig. R4: Zoomed in version of the plot in Fig. 1 (b) denoting the peaks being discussed in this section.

(f) No complete spectra, no count rates to evaluate statistical errors, no information on background subtraction routines to evaluate the systematic errors are provided.

*We thank the reviewer for pointing these out. All of these have now been **added** to the revised manuscript / Methods section. Full XEDS spectrum (as counts versus X-ray energy) corresponding to Fig. 1 (b) of manuscript is now shown in the Supplementary Information now as Fig. S1. The background subtraction is done by filtered least squares fitting.⁸*

(g) The authors seem not to have applied any absorption correction (which may be OK given the specimen is thin and the x-ray lines are all rather hard), but unfortunately two other effects they seem to have ignored could neatly explain why they observe so high Ga/Au ratios (0.45 ± 0.09 for

temperatures under 500 °C in Table S2) and As/Au ratios (0.04 ± 0.01 for the same range):

firstly, sideways scattered electrons from the heavily scattering gold dominated cap will create additional Ga K (as well as As K!) x-rays by stray excitation in the adjacent GaAs nanowire,

secondly, x-rays from Au L-lines will create Ga and As K-line x-rays by mutual fluorescence. As the position of the electron beam during EDX acquisition was not precisely recorded, the specimen drifting and growing during that time, the only conclusion will be that it cannot have been further away from the GaAs nanowire than the radius of the spherical Au-Ga-As cap, i.e. 10-20nm. This means that the Ga and As content of the catalyst cap cannot be reliably measured in this way.

We have used absorption correction during the analysis. We now mention this in the methods section.

The Ga% and As% obtained from XEDS measurements are now treated only as upper limits. We agree that there could be some unintentional scattered signal from the nanowire, though small, when we measure the catalyst composition. We have now added the following to the main text 'Scattering from the GaAs nanowire could easily give rise to this signal (further information is available in the Methods section), in which case the Ga concentration mentioned throughout the draft could be overestimated by a couple of percentage.'

While interpreting the measured Ga% to obtain the lower bound of As% from the phase diagram, we assume the measured Ga % would be higher than that given by phase diagram. Thus, our interpretation is not affected by any of the overestimation errors suggested by the reviewer.

More regarding sideward scatter from catalyst to wire:

Fig. R5: STEM line-scan of a nanowire/catalyst measured post-growth.

A STEM XEDS line scan of a nanowire/catalyst is shown in Fig. R5. It shows that there could be signal from parts that are up to about 10-15 nm away. So there could be some negligible, but non-zero signal from the nanowire part, even though we were trying to keep the beam only at the front end of the NW. If Ga% is actually less than measured, then the corresponding As from phase

diagrams will be above our estimated lower limit (since we are only estimating the lower limit on As from the phase diagram), and hence will not contradict interpretations.

Regarding mutual fluorescence:

Just like scattering, this would also overestimate the Ga% and As%, and as explained above, Ga% and As% are treated only as upper limits in this manuscript; so there is no contradictions. That said, it is extremely unlikely that Au-L x-ray photons will excite anything in the nanoparticle due to such small dimensions. The penetration depth of a 9.7 keV photon (Au-L) is in the order of a millimetre (https://en.wikipedia.org/wiki/Penetration_depth).

f) As the authors calculated correctly on page 6, ~1at% As would be required in the Au-Ga-As cap to instantaneously precipitate out a single As monolayer during the growth of the GaAs nanowire. Such monolayers are typically added more or less instantaneously during growth (see the clusters in the authors growth rate diagram in Figure 5c!), so the As content of the cap should strongly oscillate between some minimum solid solubility and a maximum ~1at% higher – this implies it cannot be lower than 1at%. However, authors estimate on page 9 that it could be in the range 0.01...0.3at% based on thermodynamic considerations, despite their EDX measurements in Table S2. At such low As content, the cap would never contain enough As to add a single monolayer to the zinc blende (or wurtzite) lattice at the speeds observed, instead atoms would have to be added one by one, replenished from the vapor phase and diffuse to the growth front where they would be incorporated to yield a gradual growth by continuous extension and completion of existing monolayers. In the phase diagram calculated and displayed as Figure 4b the measurement points could well all lie much further left, at somewhat lower Ga content but much higher As content!

*We totally agree with the logic; however the assumption of instantaneous layer formation does not agree with observations, as in fact we do observe that for all conditions discussed in this manuscript, the step-flow growth of each layer is NOT instantaneous, but takes about 0.2 - 0.5 s since each layer starts, as shown in Fig.1 in Ref. [9] from our group.⁹ We have now **added** a line in the manuscript mentioning this. Earlier reports of in situ growth of GaAs nanowires also show that the step-flow is not instantaneous.^{5,10}*

Indeed, this indicates that the actual As content during growth must always remain below 1 %. However, for the purpose of this manuscript we are not aiming to further interpret the As content from the step flow dynamics.

[Redacted]

References

1. Larsen, C. A., Li, S. H., Buchan, N. I., Stringfellow, G. B. & Brown, D. W. Kinetics of the reaction between trimethylgallium and arsine. *J. Cryst. Growth* **102**, 126–136 (1990).
2. Tornberg, M. et al. Kinetics of Au–Ga Droplet Mediated Decomposition of GaAs Nanowires. *Nano Lett.* (2019). doi:10.1021/acs.nanolett.9b00321

3. *Computational thermodynamics calphad method | Materials science. Cambridge University Press (2007).*
4. *Lehmann, S., Jacobsson, D. & Dick, K. A. Crystal phase control in GaAs nanowires: opposing trends in the Ga- and As-limited growth regimes. Nanotechnology* **26**, 301001 (2015).
5. *Jacobsson, D. et al. Interface dynamics and crystal phase switching in GaAs nanowires. Nature* **531**, 317 (2016).
6. *Walther, T. What environmental transmission electron microscopy measures and how this links to diffusivity: thermodynamics versus kinetics. J. Microsc.* **257**, 87–91 (2015).
7. *Wang, Y., He, J., Liu, C., Chong, W. H. & Chen, H. Thermodynamics versus Kinetics in Nanosynthesis. Angew. Chem. Int. Ed.* 2022–2051 (2018).
8. *Statham, P. J. Deconvolution and background subtraction by least-squares fitting with prefiltering of spectra | Analytical Chemistry. Analytical Chemistry* **49** (1977) 2149.
9. *Maliakkal, C. B. et al. Step-flow growth of III-V nanowire layers. ArXiv.190508225 Cond-Mat Physicsphysics (2019).*
10. *Harmand, J.-C. et al. Atomic Step Flow on a Nanofacet. Phys. Rev. Lett.* **121**, 166101 (2018).